# Applying Multi-Metric Deformable Image Registration for Dose Accumulation in Combined Cervical Cancer Radiotherapy

**DOI:** 10.3390/jpm13020323

**Published:** 2023-02-13

**Authors:** Qi Fu, Xin Xie, Yingjie Xu, Jing Zuo, Xi Yang, Wenlong Xia, Jusheng An, Manni Huang, Hui Yan, Jianrong Dai

**Affiliations:** 1Department of Radiation Oncology, National Cancer Center/National Clinical Research Center for Cancer/Cancer Hospital, Chinese Academy of Medial Sciences and Peking Union Medical College, Beijing 100021, China; 2Clinical Oncology School of Fujian Medical University, Fujian Cancer Hospital, Fuzhou 350014, China

**Keywords:** cervical cancer, multi-metric, deformable image registration, dose accumulation, external beam radiation therapy, brachytherapy

## Abstract

(1) Purpose: Challenges remain in dose accumulation for cervical cancer radiotherapy combined with external beam radiotherapy (EBRT) and brachytherapy (BT) as there are many large and complex organ deformations between different treatments. This study aims to improve deformable image registration (DIR) accuracy with the introduction of multi-metric objectives for dose accumulation of EBRT and BT. (2) Materials and methods: Twenty cervical cancer patients treated with EBRT (45–50 Gy/25 fractions) and high-dose-rate BT (≥20 Gy in 4 fractions) were included for DIR. The multi-metric DIR algorithm included an intensity-based metric, three contour-based metrics, and a penalty term. Nonrigid B-spine transformation was used to transform the planning CT images from EBRT to the first BT, with a six-level resolution registration strategy. To evaluate its performance, the multi-metric DIR was compared with a hybrid DIR provided by commercial software. The DIR accuracy was measured by the Dice similarity coefficient (DSC) and Hausdorff distance (HD) between deformed and reference organ contours. The accumulated maximum dose of 2 cc (D_2cc_) of the bladder and rectum was calculated and compared to simply addition of D_2cc_ from EBRT and BT (ΔD_2cc_). (3) Results: The mean DSC of all organ contours for the multi-metric DIR were significantly higher than those for the hybrid DIR (*p* ≤ 0.011). In total, 70% of patients had DSC > 0.8 using the multi-metric DIR, while 15% of patients had DSC > 0.8 using the commercial hybrid DIR. The mean ΔD_2cc_ of the bladder and rectum for the multi-metric DIR were 3.25 ± 2.29 and 3.54 ± 2.02 Gy_EQD2_, respectively, whereas those for the hybrid DIR were 2.68 ± 2.56 and 2.32 ± 3.25 Gy_EQD2_, respectively. The multi-metric DIR resulted in a much lower proportion of unrealistic D_2cc_ than the hybrid DIR (2.5% vs. 17.5%). (4) Conclusions: Compared with the commercial hybrid DIR, the introduced multi-metric DIR significantly improved the registration accuracy and resulted in a more reasonable accumulated dose distribution.

## 1. Introduction

The standard treatment for locally advanced cervical cancer is concurrent chemotherapy and radiotherapy. The radiotherapy involves external beam radiotherapy (EBRT) followed by brachytherapy (BT). Nowadays, 3D image-guided radiotherapy has been widely used in both EBRT and BT, which allows evaluating dose-volume histogram (DVH) and 3D dose distributions [1]. Ideally, the 3D dose distribution of both treatment modes should be summed to assess the total doses to target volumes and organs at risk (OARs). However, EBRT and BT are implemented independently, resulting in multiple fractional dose distributions associated with different planning images. As these images are acquired at different times and under different conditions, many factors could cause large and complex organ deformations, such as the insertion of BT applicators, varied organ filling, bowel gas, and tumor shrinkage. Therefore, image registration is difficult, making it less accurate to accumulate multi-fractional dose distributions. Currently, a common approach to assess total doses to OARs is simply adding both DVH parameters of EBRT and BT without image registration. This approach is called “worst case assumption” in the GEC ESTRO recommendations, assuming that the high-dose region is located in the same part of the OAR in each treatment. It is not always true, especially in the case of significant dose gradients from external beam boosts.

Deformable image registration (DIR) is an effective method to deal with complex registration problems. This method accounts for anatomic variations and provides a spatial transformation relationship between the volume elements of corresponding structures across different images. This nonrigid transformation can be applied to dose distributions, thereby enabling dose accumulation with high precision. The similarity metrics used for registration can be classified into two categories: Geometry-based metrics and intensity-based metrics [2]. Jamema et al. reviewed image registration in gynecological radiotherapy and showed that hybrid DIR methods with multiple metrics would result in higher registration accuracy [3]. These methods may be performed using commercial software, such as MIM Maestro (MIM Software Inc., Cleveland, OH, USA) [4,5,6,7,8], Velocity (Varian Medical Systems, Palo Alto, Santa Clara, CA, USA) [9,10,11,12], RayStation (RaySearch Laboratories, Stockholm, Sweden) [13,14], etc. [15]. A DIR algorithm involves a large number of parameters, which should be adjusted for different applications. However, in commercial software, the parameter settings are universal and mostly cannot be specified by users. This may cause undesired registration results in dealing with large and complex organ deformations. Recently, several in-house developed DIR methods demonstrated their success in registering complex deformations for combined cervical cancer radiotherapy [16,17,18,19,20,21,22,23]. However, most of these methods aimed at specified organs and lacked adequate clinical validation to be widely used. Among them, Elastix is a free, open-source software package for DIR, consisting of a collection of DIR algorithms to solve various image registration problems [24,25]. It has proven to be effective for image registration in head and neck, lung, liver, and other cancers [26,27,28,29]. We also have applied it for post-operative breast cancer radiotherapy in previous studies [30,31].

In this study, we employed a multi-metric DIR method with a penalty term provided by Elastix. The parameter setting was fine-tuned to improve the registration accuracy. The DIR performance was quantitively evaluated by comparing it to a commercial hybrid DIR method. The accumulated doses of OARs achieved by both DIR methods were analyzed.

## 2. Materials and Methods

### 2.1. Patient Data

A total of 20 cervical cancer patients (stages IB2-IIIC2r) treated with a combination of EBRT and BT were retrospectively selected for this study. EBRT was delivered to the pelvis with a total dose of 45–50 Gy in 25 fractions using volumetric-modulated arc therapy (VMAT) with 6 MV X-rays. Eleven patients received an additional simultaneous integrated boost of 10–15 Gy to involved lymph nodes in the bilateral parametria, which did not overlap the BT boost region. The EBRT plan was optimized to ensure that 100% of the prescription dose covers at least 95% of the target volume while minimizing the doses of OARs. CT-guided high-dose-rate BT was delivered using tandem/ovoid applicators and interstitial needles (if needed). The BT plan was manually optimized according to the dose constraints of OARs. The fraction dose was normalized to cover 90% of the high-risk CTV, ranging from 5 to 7 Gy.

CT scans were performed with a Brilliance CT Big Bore (Philips, Amsterdam, Netherlands) or a Somatom Definition AS 40 (Siemens Healthcare, Forchheim, Germany) with a 512 × 512 matrix. The CT slice thickness was 5 mm for EBRT and 3 mm for BT, respectively. Organs were delineated by one radiation oncologist on the treatment planning CT for both EBRT and BT. For reference, diagnostical MRI was used. The mean volume of the bladder delineated in CT images for EBRT was significantly larger than that for BT (369.75 ± 168.76 vs. 170.99 ± 103.83 cc, *p* < 0.001). While the mean volume of the rectum in CT images for EBRT was statistically smaller than that for BT (50.00 ± 20.74 vs. 56.91 ± 31.87 cc, *p* < 0.001).

### 2.2. Preprocessing

Intensity-based DIR prefers a consistent image set for higher accuracy. However, the CT numbers between the EBRT and BT images are inconsistent, such as bowel gas and the presence of BT applicator and vaginal packing. To minimize these effects, we additionally delineated the whole uterus and vagina (U + V) to include the entire applicator and vaginal packing. The CT numbers inside the bladder, rectum, and U + V contours for both CT sets were revised to 0, 30, and 60 Hounsfield units, respectively, according to their average CT numbers. Figure 1 shows a BT image after preprocessing.

### 2.3. Registration

The framework of the multi-metric DIR is shown in Figure 2. The planning CT of EBRT was set as the moving image and registered to the planning CT of the first BT fraction. Two types of similarity metrics, mutual information (MI) and kappa statistics (KS), were adopted. The *MI* is an intensity-based metric that makes use of image voxel intensity for registration. It is defined as [32]:(1)$$MI\left(\mu ;\;I_{F},\;I_{M}\right)=\underset{f\in L_{F}}{\sum }\underset{m\in L_{M}}{\sum }p\left(f,m;\mu \right)\log_{2}\left(\frac{p\left(f,m;\mu \right)}{p_{F}\left(f\right)p_{M}\left(m;\mu \right)}\right)$$
where *µ* is the vector that contains the values of transformation parameters, *L_F_* and *L_M_* are the discrete sets of intensities associated with the fixed (*I_F_*) and moving images (*I_M_*), respectively, *p* is the discrete joint probability density function, and *p_F_* and *p_M_* are obtained by summing *p* over *f* and *m*, respectively. The *KS* is a contour-based metric used to achieve the maximum overlap between the corresponding contours of paired images. The definition of the *KS* is:(2)$$KS\left(\mu ;\;I_{F},\;I_{M}\right)=\frac{2\sum_{x_{i}\in \Omega_{F}}1_{I_{F}\left(x_{i}\right)=f,I_{M}\left(T_{\mu }\left(x_{i}\right)\right)=f}}{\sum_{x_{i}\in \Omega_{F}}1_{I_{F}\left(x_{i}\right)=f}+1_{I_{M}\left(T_{\mu }\left(x_{i}\right)\right)=f}}$$
where 1 is the indicator function, and *f* is a user-defined foreground value that defaults to 1. In this study, we used the bladder, rectum, and U + V as three KS metrics for the registration. Additionally, a penalty to regularize the nonrigid transformation was used, called the transform bending energy penalty. It is defined in 2D as:(3)$$p_{BE}\left(\mu \right)=\frac{1}{P}\underset{{\tilde{x}}_{i}}{\sum }\|\frac{\partial^{2}T}{\partial x\partial x^{T}}\left({\tilde{x}}_{i}\right){\|}_{F}^{2}$$
where *P* is the number of points ${\tilde{x}}_{i}$, and the tilde denotes the difference between a variable and a given point over which a term is evaluated. As can be seen, it could penalize sharp deviations in the transformation, such as foldings. The transformation parameters can be found by minimizing a cost function to achieve the best image alignment according to the similarity metric. For this multi-metric DIR, the cost function can be formulated as:(4)$$C=\frac{w_{MI}MI+w_{B}KS_{B}+w_{R}KS_{R}+w_{U+V}KS_{U+V}+w_{P}p_{BE}}{w_{MI}+w_{B}+w_{R}+w_{U+V}+w_{P}}$$

The weights for the metrics were determined as: *w_MI_*:*w_B_:w_R_*:*w_U+V_*:*w_p_* = 20:1:1:2:1000, through the grid search method.

In terms of image transform, we used a common three-step strategy: First rigid, then affine, and finally, B-spline transformations. The B-spline transformation adopted a six-level resolution registration strategy. In each resolution level, a Gaussian pyramid was used to smooth and down-sampled the images. An advanced stochastic gradient descent method was used for optimization in each level, with a maximum iteration number of 2000. The grid spacing of the B-spline transformation in the finest resolution level for MI, bladder, rectum, and U + V was specified with 16, 12, 8, and 10 voxels, respectively. Free, open-source software, 3D Slicer, was employed for data analysis and visualization [33]. The multi-metric DIR was performed using SlicerElastix extension of 3D Slicer on an Inter^®^ Core^TM^ (Intel Core, Santa Clara, CA, USA) dual-core processor (3.0 GHz).

### 2.4. Evaluation

To evaluate the performance of the multi-metric DIR, we compared it with a hybrid DIR provided by commercial software, MIM maestro (MIM Software, Cleveland, OH, USA). The hybrid DIR is also a combination of both contour-based and intensity-based DIR. For an effective comparison, the images and contours used for the two DIR methods were the same. To prevent unrealistic deformations from occurring, the dynamic regularization was activated, and the smoothness factor was set to 0.7.

The DIR accuracy was quantitatively evaluated by computing the Dice similarity coefficient (DSC) and Hausdorff distance (HD) between the BT contour and the deformed EBRT contour in the BT image space. The DSC is used to evaluate the degree of agreement between the referenced (A) and deformed (B) contours, which is calculated by [34]:(5)$$DSC=\frac{2\left(A\cap B\right)}{A+B}$$

The *DSC* value ranges from 0 to 1, with 0 being no spatial overlap and 1 perfect agreement. The *HD* is used to measure the degree of mismatch between two contours by measuring the distance of the point of A that is farthest from any point of B and vice versa [35]. It is defined as:(6)$$HD\left(A,B\right)=max\left(h\left(A,B\right),h\left(B,A\right)\right)$$
where
(7)$$h\left(A,B\right)=\underset{a\in A}{\max}\left\{\underset{b\in B}{\min}\|a-b\|\right\}$$
and ‖·‖ is some underlying norm on the points of A and B. The statistical significance of the results was proven with a two-sided paired *t*-test at 5%-level significance.

### 2.5. Dose Accumulation

For the purpose of dose accumulation, all doses were converted into the equivalent doses in 2 Gy fraction (EQD2) using the linear quadratic model with α/β = 10 Gy for tumor and α/β = 3 Gy for normal tissues. This study aimed at the registration between EBRT and BT rather than multi-fractionated BT. For the sake of simplicity, we only used the first fractional BT dose and magnified it four times to represent the total BT dose. The deformation fields obtained from the DIR process were applied to the EBRT dose distributions for dose accumulation. According to published guidelines, minimal dose received by the most irradiated 2 cc volume (D_2cc_) of bladder and rectum were critical dosimetric parameters that correlate with an increased risk of radiation toxicity [1]. We calculated total D_2cc_ by two approaches: Simply addition of D_2cc_ values from EBRT and BT and accumulated DVH using DIR. The difference of D_2cc_ (ΔD_2cc_) between the two approaches (simply DVH addition minus dose accumulation) was analyzed.

## 3. Results

### 3.1. Registration Accuracy

The DSC and HD results of the bladder, rectum, and U + V are listed in Table 1. Compared to the hybrid DIR, the multi-metric DIR resulted in significantly higher DSC values for all contours, especially for the bladder and rectum (*p* ≤ 0.001). In total, 70% of the patients had the DSC values of all contours higher than 0.8. However, the percentage for the hybrid DIR was only 15%. The mean HD for the multi-metric DIR was also lower than that for the hybrid DIR.

Additionally, we analyzed the relationship between the registration accuracy and the volume change from EBRT to BT for the bladder and rectum, respectively. The volume ratio is defined as V_BT_/V_EBRT_. As shown in Figure 3, the DSC and HD values of both bladder and rectum for the hybrid DIR had more evident fluctuations with the changes in volume ratio. In terms of the bladder, the DSC showed a significant decline, while the HD showed a significant increase when the volume ratio was below 25%. However, for the rectum, the variation rules of both DSC and HD were not evident because the variation of the rectal volume ratio was relatively small (from 80% to 150%).

### 3.2. Dosimetric Accuracy

Figure 4 shows the results of ΔD_2cc_ results achieved by the two DIR methods. The mean ΔD_2cc_ of the bladder and rectum for the multi-metric DIR were 3.25 ± 2.29 and 3.54 ± 2.02 Gy_EQD2_, respectively, and those for the hybrid DIR were 2.68 ± 2.56 and 2.32 ± 3.25 Gy_EQD2_, respectively. As can be seen in Figure 4, the hybrid DIR had two negative ΔD_2cc_ of the bladder and five negative ΔD_2cc_ of the rectum. Obviously, these were all unreal. The proportion of unrealistic cases was 17.5%. Whereas the multi-metric DIR had only one negative ΔD_2cc_ of the bladder. The unrealistic case proportion was 2.5%. As the patient order of Figure 4b was the same as Figure 3b, it is worth noting that the unrealistic rectal D_2cc_ mainly occurred when the rectal volume ratio was above 110%. Figure 5 shows the registration results and the corresponding accumulated dose distributions for one patient using the two DIR methods.

## 4. Discussion

Dose accumulation of EBRT and BT is critical to evaluate the total biologic effective dose to targets and OARs for combined cervical cancer radiotherapy. For a more accurate accumulated dose distribution, a multi-metric DIR method based on the Elastix software was employed, which included an intensity-based metric, three contour-based metrics, and a penalty term. Reviewing previous studies, different patient data and methods used for DIR would lead to considerably different results. For example, the mean rectal DSC reported by three studies were 0.75, 0.85, and 0.92, respectively, although they used the same software for DIR [10,11,12]. Therefore, we did not compare our results with those published in other literature but rather chose a commercial hybrid DIR method that uses the same metrics as the multi-metric DIR for comparison. The results showed that the multi-metric DIR has higher accuracy than the hybrid DIR. This is mainly because the open-source Elastix software allows users to pick appropriate components and optimize parameters for a specific registration application. Through grid search, we made the parameter setting of multi-metric DIR more effective in dealing with registration tasks of large and complex organ deformations.

The metric weight is one of the most important parameters for registration. The intensity-based metric is used for global registration and may lead to a low DSC value of interested contour, whereas the contour-based metric only focuses on the specified contour surface and may cause unrealistic deformations. There is a need to find a balance between the weights of the two types of metrics. With the grid search method, the optimal weights of the two metrics are 20:1, and the metric weights of the bladder, rectum, and U + V are 1:1:2. The six-level resolution registration is an efficient strategy to deal with large deformations. The grid spacing in the last resolution level is also an important parameter that affects registration accuracy. A large grid spacing can match large structures but skip small structures. On the other hand, a small grid spacing may improve the accuracy but may also cause unrealistic deformations. The final grid spacing for bladder, rectum, and U + V (12:8:10 in voxel) were set according to their volumes.

After reviewing the registration results, we found that volume shrinkage of the bladder and rectum from EBRT to BT would cause foldings. To minimize this influence, the transform bending energy penalty was added to the cost function. In our experience, the relative weight of the penalty ranging from 100 to 1000 could effectively reduce foldings while having little impact on the DSC values. For the same purpose, the hybrid DIR used the smoothness factor and activated dynamic regularization. However, their effect is less. There were still a lot of foldings, while the DSC values were also impacted when the smoothness factor was set to 0.7.

The main cause of the unrealistic deformations is the volumetric change of organs between the CT sets. As can be seen in Figure 3, the larger the volumetric change, the more difficult the registration task, and the more evident the fluctuations of the DSC values. The fluctuations were especially larger for the hybrid DIR. Furthermore, the unrealistic rectal D_2cc_ mainly occurred when the rectal volume ratio was above 110%. These suggested that the volumetric changes of the bladder and rectum should be minimized if applying DIR in clinical registration tasks. For example, a Foley catheter could be placed in the bladder before each treatment. Nevertheless, the multi-metric DIR still resulted in 70% of the patients with all DSC values reaching the requirement given by AAPM TG 132 report (0.8–0.9). Therefore, it is reasonable to believe that the multi-metric DIR could be feasible if the volumetric changes were well controlled. One drawback is that the multi-metric DIR takes a relatively long time for preprocessing and optimization. Usually, the optimization process needs to take a few minutes to obtain an optimal result. By contrast, the hybrid DIR only takes less than a minute. This suggests that the efficiency of multi-metric DIR should be further improved in the future.

The limitation of this study is that it only focused on the registration between EBRT and BT. Because most patients in our hospital are not treated with CT-guided BT in each fraction, we did not discuss the inter-fraction variations during BT treatment. According to the study of Jamema et al., the spatial location of the D_2cc_ region of bladder and rectum is quite stable for each BT fraction [36]. Thus, we believe it is reasonable to use the magnified BT dose of the first fraction to represent the total D_2cc_ of bladder and rectum from BT, although it is not the actual total BT dose.

However, it should be pointed out that although the accuracy of the multi-metric DIR was significantly improved, it still could not prevent unrealistic dose accumulation from occurring. Moreover, there is no relationship found between the DSC values and the unrealistic dosimetric results, indicating that the DSC is not sufficient to evaluate the accuracy of dose accumulation. Chetty et al. reviewed several approaches to estimate the impact of DIR uncertainties on dose mapping [37]. However, at present, there is a lack of straightforward measures to qualitatively or quantitatively assess the accuracy of the deformed dose distributions. Considering that DIR may cause various uncertainties and the ΔD_2cc_ is only around 3 Gy, DVH addition is still a reliable approach to assess the total doses of targets and OARs, as recommended in refs. [3,38,39]. The multi-metric DIR requires further development for accuracy and efficiency before being ready for clinical application.

## 5. Conclusions

In this study, we introduced a multi-metric DIR method based on open-source Elastix software. Compared to the commercial hybrid DIR, the multi-metric DIR has higher accuracy and can achieve a more reasonable accumulated dose distribution, suggesting that it has potential for clinical application.

## Figures and Tables

**Figure 1 jpm-13-00323-f001:**
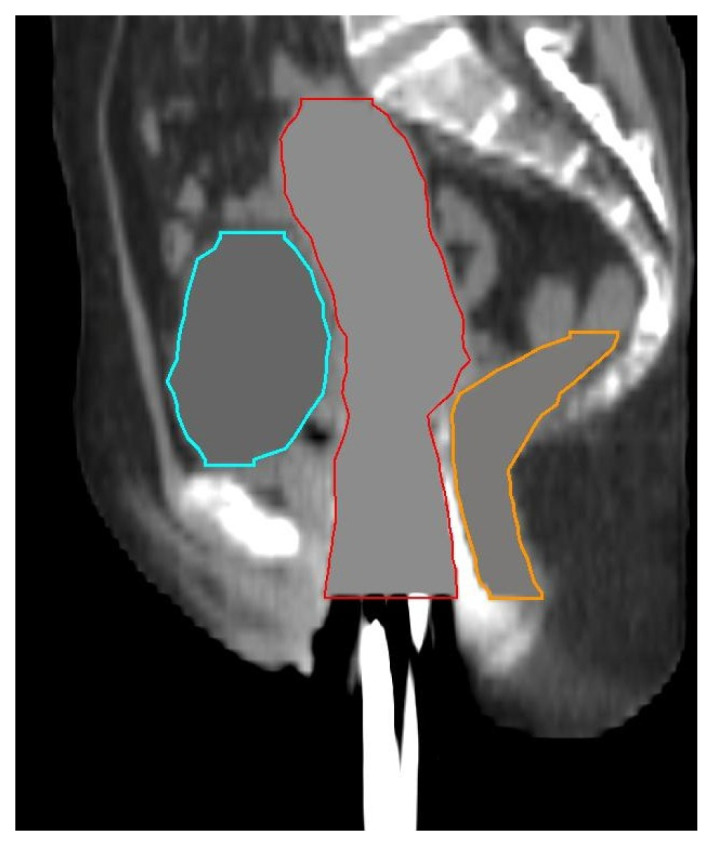
An example of sagittal BT image after preprocessing. The cyan, orange, and red contours represent bladder, rectum, and U + V, respectively.

**Figure 2 jpm-13-00323-f002:**
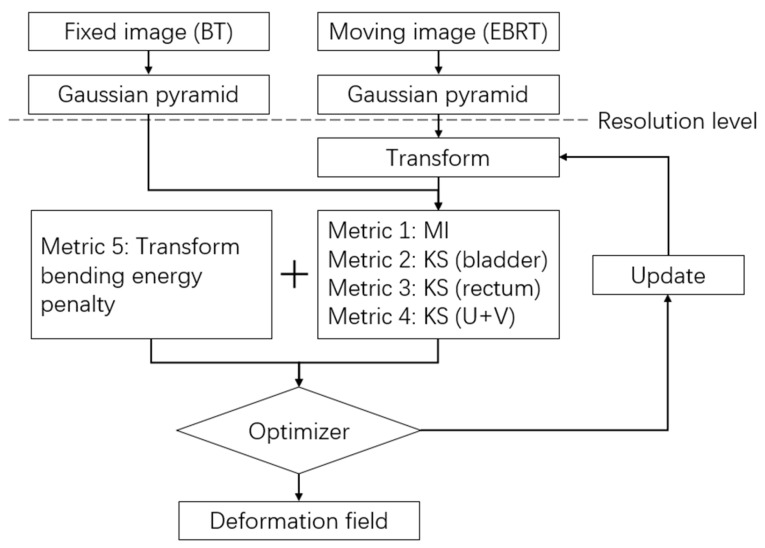
Flowchart of the multi-metric DIR process.

**Figure 3 jpm-13-00323-f003:**
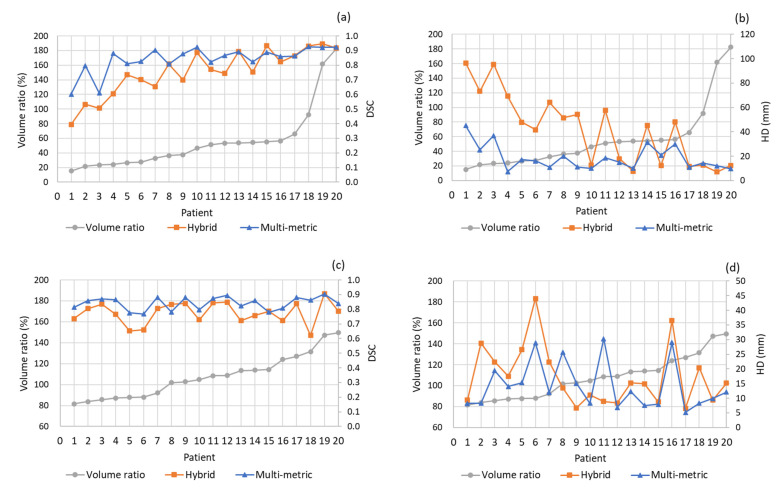
The changes of DSC (**a**) and HD (**b**) values with respect to the volume ratio of BT to EBRT for bladder and those (**c**,**d**) for rectum. Note that the orders of the patients between (**a**,**b**) and (**c**,**d**) are different.

**Figure 4 jpm-13-00323-f004:**
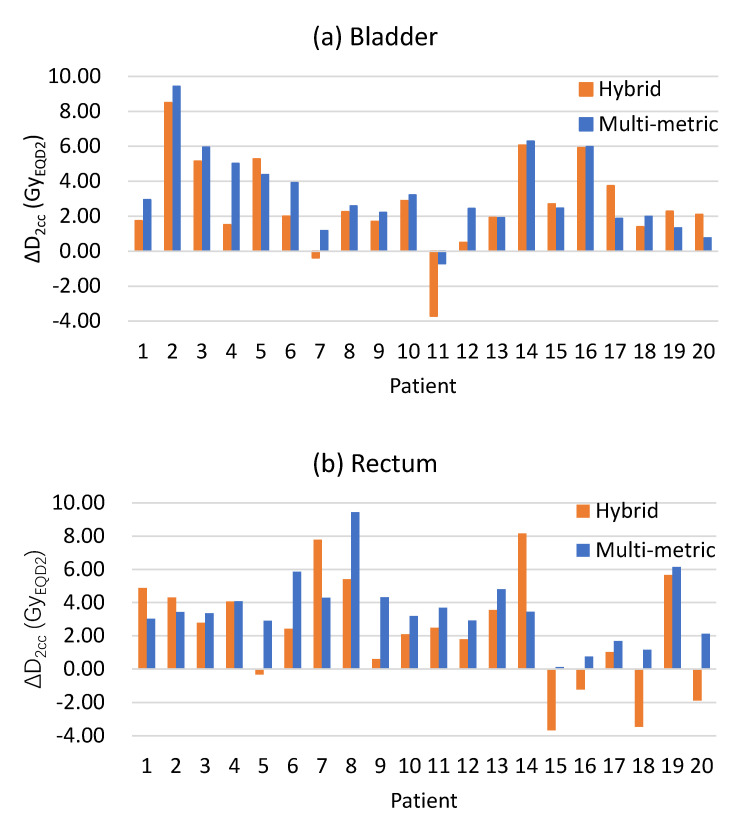
ΔD_2cc_ of (**a**) bladder and (**b**) rectum achieved by both multi-metric and hybrid DIR methods. The orders of the patients in (**a**,**b**) are consistent with those in Figure 3a–d, respectively.

**Figure 5 jpm-13-00323-f005:**
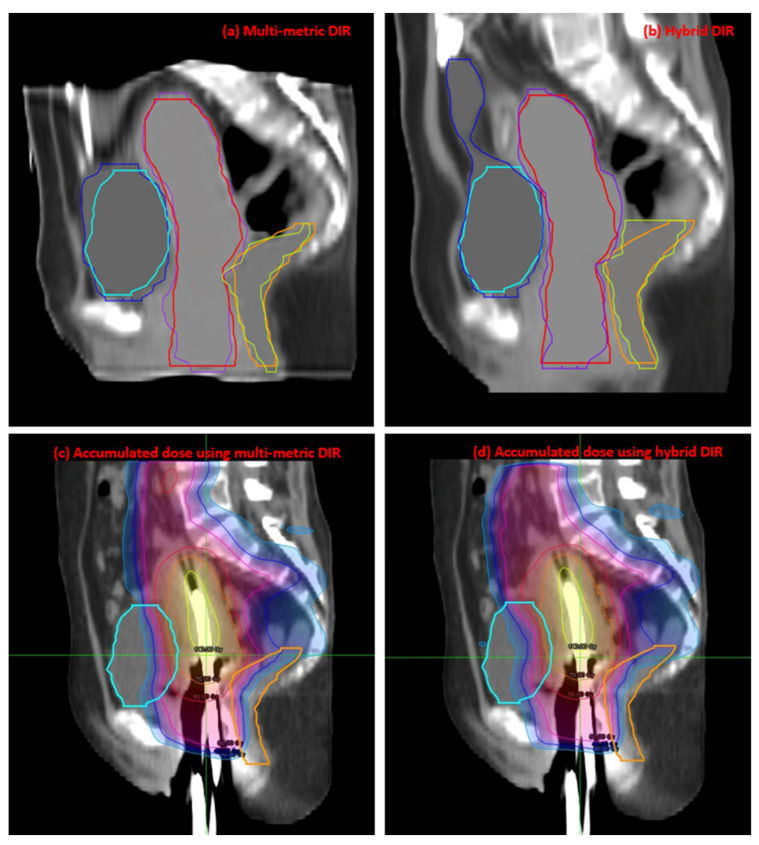
The registration results (**a**,**b**) achieved by the multi-metric and hybrid DIR methods, respectively, and their corresponding accumulated dose distributions (**c**,**d**) for one patient. The reference bladder, rectum, and U + V are shown in cyan, orange, and red, respectively, and the corresponding deformed contours are shown in blue, yellow, and pink, respectively.

**Table 1 jpm-13-00323-t001:** DSC and HD of bladder, rectum, and U + V between multi-metric and hybrid DIR (mean ± standard deviation).

Contour	Parameters	Multi-Metric	Hybrid	*p*-Value
Bladder	DSC	0.84 ± 0.09	0.76 ± 0.15	0.001
	HD (mm)	18.68 ± 9.99	41.98 ± 28.27	<0.001
Rectum	DSC	0.84 ± 0.04	0.77 ± 0.07	<0.001
	HD (mm)	14.23 ± 7.89	17.35 ± 10.01	0.155
U + V	DSC	0.89 ± 0.02	0.86 ± 0.05	0.011
	HD (mm)	16.92 ± 5.92	17.56 ± 9.85	0.756

## Data Availability

Not applicable.

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
