# Peer review of "Applying Multi-Metric Deformable Image Registration for Dose Accumulation in Combined Cervical Cancer Radiotherapy"

_jpm, 2023, doi:10.3390/jpm13020323_

Round 1

Reviewer 1 Report

I have added my suggestions in the comment section of the article. If the authors make necessary changes in the article, it can be considered for publication.

Sincerely,

Kushal Gandhi.

Author Response

We sincerely thank reviewers and editors for reviewing our manuscript. We have revised the manuscript according to the comments. All changes have been marked up using the “Track Changes” function in the revision. The response to each comment is as follows:

Response to Reviewer 1 comments:

  1. We have added a reference at the end of this sentence as Ref. 1. Thanks for your reminder.
  2. This is a retrospective and non-interventional study, which did not adversely affect the rights and welfare of the subjects. The results were presented as statistical data and did not contain any identifiable patient information. Generally, this kind of studies do not need to apply for ethical approval and could be exempt from written informed consent. Thus, we have no IRB protocol number for this study. For the sake of clarity, we have modified this sentence as “A total of 20 cervical cancer patients …… were retrospectively selected for this study”.

If the IRB protocol number is a must according to the journal's regulations, we would immediately supplement an application to research ethics committee. It may take about one month to get an IRB protocol number.

Reviewer 2 Report

The objectives of the paper are clear and reasonable, but the paper is missing a reference method to evaluate practical value and correctness of all performed transformations and offered approaches. Proposed by the authors multi-metric transformation is compared with hybrid transformation and discrepancies can be observed in the results for these two tools.  A third reference method is necessary to make a decision which method is more accurate and valid.  Also dose evaluation performed with DIR should compared with some biological/therapeutic or dosimetric  results to show their correctness and practical value.

Some technical issues can also be noted:  

1.       Criteria (i.e. DSC and HD)  of  accuracy and correctness of all described transformations are not clearly described and explained.

2.       Diagrams on Fig  3  are hardly understandable and difficult to percept. If the aim of these diagrams to show dependance of DIR accuracy from rectum/bladder volume change from EBRT to BT then corresponding diagrams DSC vs volume ratio and HD vs volume ratio should be plotted.

3.       The meaning of  “The maximum dose to 2 cc (D2cc)” value should be described in details and its selection should be explained. So for it is hard to understand the correctness of its usage.

4.       Units in Fig 4 diagram are missing

Author Response

We sincerely thank reviewers and editors for reviewing our manuscript. We have revised the manuscript according to the comments. All changes have been marked up using the “Track Changes” function in the revision. The response to each comment is as follows:

 Response to Reviewer 2 comments

The objectives of the paper are clear and reasonable, but the paper is missing a reference method to evaluate practical value and correctness of all performed transformations and offered approaches. Proposed by the authors multi-metric transformation is compared with hybrid transformation and discrepancies can be observed in the results for these two tools. A third reference method is necessary to make a decision which method is more accurate and valid. Also dose evaluation performed with DIR should compared with some biological/therapeutic or dosimetric results to show their correctness and practical value.

Thanks for your comment. The methods we used to quantitatively evaluate the DIR quality mainly referenced the AAPM TG-132 report (Ref. 2). It described five quantitative measures of DIR accuracy, including Dice similarity coefficient (DSC), mean distance to agreement (MDA), Jacobian determinant, target registration error (TRE) and consistency. The DSC is the most common measure and is also most appropriate to be used for our study. The Hausdorff distance (HD) is another common measure used in image registration studies, such as Ref. 8, 14, 23. So, we used the DSC and HD to evaluate the accuracy of the two DIR methods. For the sake of clarity, we added the detailed explanations of these two measures. Please also see the response to Comment #1.

Actually, there is a lack of straightforward measures to qualitatively or quantitatively assess the accuracy of the deformed dose distributions. As a critical dosimetric parameter for cervical cancer, the D2cc of bladder and rectum were used for dose evaluation in most DIR studies (reviewed by Ref. 3). In Section 3.2, we also used the D2cc as dosimetric results to show different DIR performances. Please also see the response to Comment #3.

Some technical issues can also be noted: 

  1. Criteria (i.e. DSC and HD) of accuracy and correctness of all described transformations are not clearly described and explained.

Thanks for your reminder. In Section 2.4, we have added the definitions and explanations of the DSC and HD, as well as their corresponding references (Ref. 34 and 35).

  1. Diagrams on Fig. 3 are hardly understandable and difficult to percept. If the aim of these diagrams to show dependance of DIR accuracy from rectum/bladder volume change from EBRT to BT then corresponding diagrams DSC vs volume ratio and HD vs volume ratio should be plotted.

As you request, we have separately divided Fig. 3 (a) and (b) into two charts. In the revised manuscript, Fig. 3 (a) and (c) showed the DSC vs volume ratio, and Fig. 3 (b) and (d) showed the HD vs volume ratio. Thanks for your suggestion.

  1. The meaning of “The maximum dose to 2 cc (D2cc)” value should be described in details and its selection should be explained. So for it is hard to understand the correctness of its usage.

Thanks for pointing out this issue. In Section 2.5, we have modified the explanation of D2c.c as: “According to published guidelines …… increased risk of radiation toxicity”.

  1. Units in Fig 4 diagram are missing

Thanks for your reminder. In Fig. 4, the unit of ΔD2cc is GyEQD2. As mentioned in Section 2.5, it means that all doses have been converted into the equivalent doses in 2 Gy fraction (EQD2) so that they can be accumulated and compared.